# Seismic Performance and Engineering Application Investigation of a New Alternative Retainer

**DOI:** 10.3390/polym14173506

**Published:** 2022-08-26

**Authors:** Lei Yan, Guo Li, Xiaoying Gou, Ping Zhang, Xinyong Wang, Yu Jiang

**Affiliations:** 1School of Civil Engineering, Chongqing Three Gorges University, Chongqing 404100, China; 2School of Civil Engineering, Chongqing Jiaotong University, Chongqing 400060, China

**Keywords:** alternative, elastoplastic energy, retainer, fiber-reinforced polymer concrete

## Abstract

Focusing on the dilemma that the traditional lateral shear keys are ineffectual in limiting the displacement and repair of small-to-medium spanning highway bridges, this paper briefly describes the necessity of considering fiber-reinforced polymer concrete with the shear keys design, and studies the seismic performance of an alternative retainer that focuses on three functions of “limiting displacement”, “energy consumption”, and “alternative link”. In order to study the anti-seismic effectiveness under the seismic loads, four alternative retainer specimens with different sizes were designed. The quasi-static tests were carried out on four specimens, respectively. The seismic damage mode of the quasi-static alternative retainer was investigated. We examined the influence of the designed parameter of the alternative retainer on the anti-seismic effectiveness of the alternative retainer. Taking a two-span simply supported girder bridge, for example, the comparison between the seismic response of the bridge with retainers and without is analyzed based on a consideration of the sliding plate rubber bearings and the test results of the new retainers. The results show that the failure mode of the new alternative retainers is a two-stage process involving the alternative links: firstly shear failure and then the overall retainer damages, which is convenient to retrofit and reinforce post-earthquake. The thickness of the web of the alternative link, as a sensitive factor, influences the bearing capacity of the new retainers, yield displacement, ultimate displacement, ductility coefficient and overall energy consumption. The height of the alternative link will merely influence the ultimate bearing capacity, and transverse replacement of the main girder with the new alternative retainers is greatly reduced compared to without retainers, and the seismic response increase in the pier is gentle.

## 1. Introduction

History has proved that, due to the ease of sliding between the girder bottom and the plate rubber bearing, the damage of the retainer of small-to-medium spanning bridges at home and abroad is common after earthquakes. During the Wenchuan earthquake, the failure rate of the shear key was much higher than that of piers and columns, and a large number of plate rubber bearings were completely destroyed [1], thus, the dilemma that the regular lateral shear keys were ineffectual in limiting displacement and repair aroused the attention of scholars around the world.

For this, Silva, Kottariet al. [2,3], and Xu Lueqin, Han Qiang et al. [4,5] carried out concrete retainer tests to respectively investigate the influence of various parameters on their seismic performance and attempts to process these results. Meanwhile, retainers of various structural forms have sprung up one after another. Wang Kehai et al. [6] designed a new seismic double-layer stopper and recommended the implementation of a “multilevel fortification, hierarchical energy-consumption“ philosophy based on the characteristics of weak inner block and stronger outer block. Guan Z G et al. [7] proposed adding a damping and stiffness device that has stable hysteretic behavior and good fatigue characteristics, which has attracted widespread attention. Li Jianzhong [8,9] found that the improved X-type elastic–plastic steel block has high energy dissipation capacity and can effectively control the relative displacement through its experiments, and pointed out progress on the bridge seismic design of blocks that aim to be gradually transformed from seismic alleviation to post-earthquake structural resilience through summarizing the status of the theory and technology in seismic design for blocks. Xiang Nailiang et al. [10] proposed a new type of friction retainer and found that, although it can effectively reduce the displacement between pier and girder, it may increase the seismic damage of the substructure by carrying out a comparative test with conventional concrete retainers. Existing research has confirmed that reasonable block setting will effectively reduce the seismic response of the bridge. Furthermore, the application of “structural fuse elements” and the principle named “one-can and three-easy” was proposed, which means, in devastating earthquakes, “the injury component and extent of injury can be controlled, the injury position is easy to check, the injury component is easy to repair, the damaged component is easy to change”, has also been widely recognized [11,12]. Even so, various types of blocks are rarely able to truly implement the above design concepts because of the limitations of conventional concrete materials and incomplete structural optimization, the repair and reinforcement requirements of the blocks after earthquakes are especially difficult to meet. Fortunately, the introduction of fiber-reinforced polymer concrete (FRP) has injected new vitality into the retainer design. Fiber-reinforced polymer concrete (FRP), a new cement composite, is generally composed of high-strength continuous fibers with a polymer matrix that acts as a binder and has excellent crack resistance and mechanical properties which has attracted many scholars in various fields [13]. Billah et al. [14] used carbon fiber composite material (CFRP) to reinforce the multi-column bent pier, and found that CFRP reinforcement can effectively improve the bending performance of the bent pier through the seismic test. Algin et al. [15] proposed that the concrete material with basalt fiber reduced the connected pores and bleeding channel of material and improved the durability of the concrete. Jony et al. [16] pointed out that the cracks in fiber-reinforced polymer composites (FRPC) can self-heal in a variety of ways and proposed a self-healing agent to prolong the blocks’ life. Chen guangming et al. [17] studied the engineering application effect of a new type of FRP–concrete–steel double-wall hollow pier composite structure, and found that it not only gives full play to the characteristics of the material to greatly improve the seismic performance, it also reduces the preparation cost. Therefore, it provides new opportunities for the new blocks with the new structure of fiber-reinforced polymer concrete. Dolati et al. [18] proposed and investigated a novel method for splicing prestressed precast concrete piles based the excellent mechanical performance of fiber-reinforced polymer concrete (FPR), and found the construction and repair time were greatly shortened, which acts as a positive reference for fiber-reinforced polymer concrete blocks. Wu wen-peng et al. [19] found that merely improving the bearing capacity of the blocks plays a small role in shortening the repair time of bridges post-earthquake, the new blocks made full use of the material properties of high strength concrete after incorporating the idea of assembled design, which promoted its overall performance and was in favor of its post-earthquake retrofit and reinforcement, with fiber-reinforced polymer concrete block design as a reference. It can clearly be seen that fiber-reinforced polymer concrete has been widely used in various fields due to its excellent material properties for blocks, in addition, fiber-reinforced polymer concrete has a good foundation and technical reserves, which is beneficial to the realization of advanced design concepts of blocks. In conclusion, fiber-reinforced polymer concrete is worth investigating, and it must be noticed that the excellent properties of fiber-reinforced polymer concrete materials can be sufficiently utilized only by matching with the reasonable structural design.

Therefore, the paper studied a new alternative retainer [20] based on the application of “structural fuse elements” and the principle named “one-can and three-easy”, and focused on three functions of “displacement control”, “energy consumption”, and “alternative link”. The paper also analyzed the anti-seismic performance of the retainer through a quasi-static test and numerical simulation of a bridge model to lay the foundation for the integration of the retainer design and fiber-reinforced polymer concrete performance in the future.

## 2. The Quasi-Static Test

### 2.1. Structure Design of New Retainer

The composition of the alternative retainer is shown in Figure 1, which includes five parts: the vertical plate, the alternative link, the upper top plate, the lower bottom plate, and the adjustable connecting side plate. The installation diagram is shown in Figure 2. The alternative link, located between the vertical plates, is connected by high-strength bolts, the main structure is fixedly connected with the beam body and the bridge bent cap through embedded bolts and steel bars, etc. The working mechanism of the alternative retainer is as follows: in the event of frequent earthquakes, the alternative link undergoes yield failure due to shear deformation for reducing the seismic response, and then the function of the entire retainer can be restored by replacing the alternative link post-earthquake; in rare earthquakes, the alternative retainer and the entire block together undergo plastic deformation to dissipate seismic energy for reducing the seismic response of the bridge and preventing girder falling. Furthermore, the energy dissipation capacity of the retainer can be adjusted by controlling the stiffness of the vertical plate of the retainer and replacing the relevant parameters of the energy-consuming link in actual engineering.

### 2.2. Experiment Overview

As shown in Table 1, the paper divided the test into four groups according to the variables of the web thickness, the alternative link height, and the installation position of the alternative link, and a total of four specimens. All of specimens underwent a low-cycle cyclic reciprocating loading test. The overall structure of the specimen is shown in Figure 3. The production process of retainer specimens is as follows: each part of the main body of specimens is formed by steel plates of different thicknesses being cut and then welded. Specifically, the alternative link is welded into a whole and then is connected by the vertical plate of the test piece through high-strength bolts.

Specimen 1 is a standard specimen, and its specific dimensions are shown in Figure 4 below.

### 2.3. Loading Protocols and Failure Criterion

The test setup is shown in Figure 5. The test and numerical analysis are carried out according to the regulations of the loading method-adopted force–displacement dual control [21], as shown in Figure 6. Specifically, before the model yields, multi-stage loading is carried out at 25%, 50%, and 75% of the estimated yield load, and each stage loading is repeated once. After the model enters the yield state, loading protocols adopt the displacement-controlled loading method through the cyclic loading manner of ±Δy, ±2Δy, ±3Δy, ±4Δy…, and each stage displacement loading is cycled three times after yielding. The model can be considered to be damaged when one of the following conditions occurs: (1) the bearing capacity decreased to 85% of the peak bearing capacity; or (2) the alternative link of the specimen is damaged due to the damage of the vertical plate, and forming a plastic rotating mechanism.

## 3. Experiment Analysis

### 3.1. Analysis of Test Phenomenon

Taking the standard specimen as an example, the initial gap between the loading surface of the specimen and the hydraulic servo test device is eliminated by preloading before the formal loading. The preloading target value is F = 50 kN, and it is unloaded after reaching the target value, which needs to repeat three times. When the preloading is completed, the loading force is started, and unloaded after reaching the target value at each level that is, respectively, F = 100 kN, F = 200 kN and F = 300 kN. The loading target value of each stage is cycled one time under the action of force loading, and the transverse displacement of the specimen is recorded after the force loading is completed, and then the displacement control loading is started.

When the target value F = 100 kN, the specimen displays no obvious deformation. When the target value F = 200 kN, the specimen has a small deformation, and each vertical steel plate of the specimen displays synergistic deformation. When the target value F = 300 kN, the inter-story drift is Δy = 18 mm.

Therefore, loading displacement on the first time is Δy = 18 mm, after that, the target value of the displacement control is Δy = 36 mm, 54 mm, 72 mm, 90 mm, 108 mm, 126 mm, and 144 mm, in turn, and the above displacement loading is cycled three times per stage.

When Δ = 18 mm, specimen 1 does not receive any new damage under the displacement control loading after three times; and then continues to use the displacement loading method. When Δ = 72 mm, the large up-and-down displacement of the alternative link leads to the serious deformation of its web and flange, and then the enters the yield state. When reaching Δ = 108 mm, the phenomenon occurs in which the vertical plate of the device is seriously distorted, the bolts of the alternative link fall off, and the weld between the side steel plate and the top plate is damaged, resulting in the decrease in the rigidity and the loss of bearing capacity of the whole specimen.

During the above loading process, the deformation of specimen 1 is shown in Figure 7, Figure 8 and Figure 9 below.

### 3.2. Analysis of Test Results

#### 3.2.1. Bearing Capacity

The average yield bearing capacity of the four specimens is 399.5 KN, and the average ultimate bearing capacity is 490.5 KN. The yield bearing capacity is increased by 24.20% and the ultimate bearing capacity is increased by 9.59% in specimen 2 compared with specimen 1. The yield bearing capacity is increased by 3.19% and the ultimate bearing capacity is increased by 7.89% in specimen 3 compared with specimen 1. The yield bearing capacity is decreased by 2.45% and the ultimate bearing capacity is increased by 0.85% in specimen 4 compared with specimen 1. It can be seen that all specimens have good bearing capacity. The final failure form of the new retainer (Figure 9) avoids the brittle shear failure of the inclined section of the traditional concrete retainer (Figure 10), which reduces the probability of damage to the bent cap and abutment cap. The data show that the yield bearing capacity and ultimate bearing capacity of the retainer will be significantly improved when the thickness of the web of the alternative link of the new block is increased; and increasing the height of the alternative link will improve the ultimate bearing capacity and have little effect on the yield bearing capacity. Furthermore, the installation position of the alternative link does not affect the ultimate bearing capacity, but prompting the installation position slightly reduces the yield bearing capacity.The bearing capacity test results of each specimen are shown in Figure 11.

#### 3.2.2. Ductility

The coefficient of ductility, an indicator describing the structural ductility capacity, characterizes the energy dissipation and deformation capacity of the specimen. Δ*_u_* and Δ*_y_* are the yield displacement and limit displacement of the model, respectively, in the formula. In this paper, a simplified method named “Farthest Point Method” [22] is used to determine the yield displacement Δ*_y_* of the specimen. The formula calculates the coefficient of ductility *μ* as follows:(1)$$\mu =\frac{\Delta_{u}}{\Delta_{y}}$$

The results of coefficient of ductility of each specimen are shown in Figure 12 below.

The average yield displacement of the four specimens is 10.1 mm, the average ultimate displacement is 104.6 mm, and the average ductility coefficient is 10.35. The yield displacement is increased by 20.41% and the ultimate displacement is increased by 30.50% in specimen 2 compared with specimen 1. The yield displacement remains the same and the ultimate displacement is increased by 30.50% in specimen 3 compared with specimen 1. The yield displacement is decreased by 15.87% and the ultimate displacement is increased by 1.50% in specimen 4 compared with specimen 1. The data show that the yield displacement, ultimate displacement and ductility coefficient of the retainer will be significantly improved when the thickness of the web of the alternative link of the new block is increased; and increasing the height of the alternative link have little effect on the yield displacement and the ultimate displacement. Furthermore, the installation position of the alternative link will decrease the yield displacement and the ultimate displacement.

#### 3.2.3. The Energy Consumption

The energy dissipation capacity of structural specimens can be measured by the area enclosed by the load-displacement hysteretic behavior. In this paper, the cumulative energy consumption EAD and the equivalent viscous damping coefficient he is used to characterize the energy dissipation capacity of the retainer, as shown in Figure 13. The formula of he is as follows:(2)$$E_{AD}=\sum_{i=1}^{n}\sum_{j=1}^{m}\Delta W_{ij}$$
(3)$$h_{e}=\frac{1}{2\pi }\frac{\left(S_{ABC}+S_{ADC}\right)}{\left(S_{BOE}+S_{DOF}\right)}$$

EAD—the cumulative hysteretic energy consumption of structural specimens; *i*, *j*—the total number of loading levels and the number of cycles loaded per level; Δ*Wij*—hysteretic energy dissipation when the cycles is *j* and the load level is *i*.

The energy consumption capacity of each specimen is shown in Figure 14 below.

The data show that the overall energy consumption capacity of the retainer will be significantly improved when the thickness of the web of the alternative link of the new block is increased; and the height of the alternative link have little effect on the energy consumption capacity. Moreover, the higher installation position of the alternative link will decrease the energy consumption capacity of the retainer.

In conclusion, it can be seen from the test results that the new alternative retainer has a stable force–displacement relationship and can complete a displacement cycle of 5Δy. The specimen not only has a high bearing capacity, but also has a good energy dissipation capacity. Furthermore, by comparing the new type of retainer failure mode with the traditional concrete failure mode after the test, it can be found that merely increasing the strength of the retainer without changing the structural form of the retainer is not beneficial to the retrofit of bridge function of post-earthquake and not changes the overall damage status between the retainer and the bent cap.

## 4. Bridge Model Analysis

### 4.1. The Establishment of Finite Element Model

Take a 2 × 20 m prestressed concrete simply supported T-beam bridge as the prototype bridge, as shown in Figure 15, with 7 T-beams. The beam height is 1.5 m, the bridge width is 16.25 m, and the substructure pier is a double-column solid pier with a diameter of 1.2 m, a pier height of 10 m, and a bent cap height of 1.6 m.

In order to simulate the structural response of the bridge using the alternative retainer under earthquake, the finite element model of the two-span simply supported girder bridge with the alternative retainer was established by ANSYS, as shown in Figure 16. The main body of the finite element model (girder, the bent cap, the pier) adopts Beam188 beam element, the elastic modulus of concrete material. $E=3.5\times 10^{4}N{/\mathrm{mm}}^{2}$, mass density $\rho =2500kg/m^{3}$, and Poisson’s ratio v=0.2.

### 4.2. Approach Implementation with ANSYS

#### 4.2.1. Sensitivity Analysis

This paper adopts a probabilistic approach to sensitivity analysis [23]. This approach is made using the commercial FEM-based package ANSYS. Advantage is taken of the ANSYS probabilistic design system (APDS) and ANSYS parametric design language (APDL). On the one hand, this enables us to use the ANSYS random generator to obtain random samples of input parameters and analyses output probability. On the other hand, it permits obtaining the sensitivities solely using the postprocessor. The principle is as follows:

Firstly, the vector a, $a=\left\{A_{1},C_{1},F_{1},K_{1}\ldots ,b_{n}\right\}$, $A_{1},C_{1},F_{1},K_{1}\ldots ,b_{n}$ is defined as the parameters of model, which are independently and arbitrarily distributed. The random parameters are defined as $a_{k}=\left\{a_{k1},\;a_{k2},\;a_{k3},\ldots ,\;a_{kn}\right\}$, and $n_{k}\in \langle 0,1\rangle$ is a group of the standard uniformly distributed random number. The cumulative distribution function $F_{a}(a_{k})$ is obtained by submitting into $n_{k}$, assuming $n_{kj}=s_{j}p^{-1}$. According to the recursion relation:(4)$$s_{j}=bs_{j}{}_{-1}+c-k_{j}{}_{-1}p$$
(5)$$a_{kj}=F_{a}^{-1}(n_{kj})$$

b and c are positive integer, $k_{j}{}_{-1}$ is the integer part of $(bs_{j}{}_{-1}+c)m^{-1}$, $s_{j}$, $s_{j}{}_{-1}$ are current and previous seed values of the recursion and j=1,2,…p.

The resulting input variable $a_{k}$ is aggregated to obtain the statistic:(6)$$\langle a_{k}\rangle =E\left[a_{k}\right]=\sum_{j=1}^{p}a_{kj}p^{-1}$$

The standard deviations:(7)$$\sigma_{x,iv}=\sqrt{V_{br}(a_{k})}=\sqrt{\sum_{j=1}^{p}(a_{kj}-\langle a_{k}\rangle {)}^{2}(p-1{)}^{-1}}$$

$V_{br}(a_{k})$ is the variance evaluation of input variable $a_{k}$.

Secondly, the output parameter $N\left[a_{k}\right]=\left[N_{k1}\left[a_{k}\right],N_{k2}\left[a_{k}\right],\ldots ,N_{kj}\left[a_{k}\right]\right]$ is obtained by resolving all of value of $a_{k}=\left\{a_{k1},\;a_{k2},\;a_{k3},\ldots ,\;a_{kn}\right\}$. The aggregating statistic $N\left[a_{k}\right]$ is likely obtained in this way.

The degree of correlation of output parameter and input parameter is measured by the coefficient of correlation r.
(8)$$r=\frac{Cov(a_{k},N\left[a_{k}\right])}{\sqrt{Vbr(a_{k})Vbr(N\left[a_{k}\right])}}$$

$Cov(a_{k},N\left[a_{k}\right])$ is the covariance estimator of input parameter $a_{k}$ and output parameter $N\left[a_{k}\right]$.

Finally, r∈(−1,1), if r=1, the relation of input parameter $a_{k}$ and output parameter $N\left[a_{k}\right]$ is completely direct dependent; if r=0, the relation of input parameter $a_{k}$ and output parameter $N\left[a_{k}\right]$ is completely independent.

As shown in Figure 17, the sensitivity parameters of finite element models are obtained in this way, which is used to filter the research problem. The sensitivity parameters support stiffness K, support the friction coefficient C, retainer yield bearing capacity $F_{1}$, ultimate bearing capacity $F_{2}$, PGA and the frequency-spectrum characteristics of seismic wave.

#### 4.2.2. Implementation of Sensitive Parameters

The simplified mechanical model of seismic isolation bearing is used to simulate the actual status of the bridge to achieve the effect of approximate simulation because there is no direct seismic isolation bearing element in ANSYS. First, in order to calculate the restoring force of the plate rubber bearing, it is assumed that the plate rubber bearing connects *I* and *j* in the finite element model of the bridge to establish a local coordinate system in which the vertical direction is the x direction. Therefore, the nodal displacements are $u_{i}$, $v_{i}$, $w_{i}$, $u_{j}$, $v_{j}$ and $w_{j}$ in the local coordinate system of node *i* and *j*, assuming $u_{i}=u_{j}$, which are processed by the method of master–slave degrees of freedom. The shear displacement of the plate rubber bearing is represented by subtraction between $u_{i}$, $v_{i}$, $w_{i}$, $u_{j}$, $v_{j}$ and $w_{j}$. Moreover, the restoring force of bearings in the y-direction and z-direction are:(9)$$\left\{ \begin{array}{l} f_{s1}=K\cdot (v_{i}-v_{j})\\ f_{s2}=K\cdot (w_{i}-w_{j}) \end{array}\right.$$
(10)$$K=\frac{GA}{\sum t}$$

K is the equivalent shear stiffness of bearings. G is the dynamic shear modulus of bearings. A is the shear area of bearings. ∑t is the total thickness of each layer of rubber sheet.

Secondly, it is reasonable that adopting Combin14 spring element as the vertical joint between the girder and the bent cap based that the vertical bearing capacity of the plate rubber bearing is mainly provided by the rubber and the multi-layer thin steel plate added inside. The mechanical principle of the diagram is shown in Figure 18. Since there is no need to consider the vertical damping effect, it is only necessary to set the spring stiffness and specify that its direction is vertical.

Thirdly, the plate rubber bearing will first undergo shear deformation during the earthquake. Because its hysteretic curve is long and narrow, its energy dissipation capacity is inferior, and the relationship between the force and the displacement of the bearing appears to be a linear relation; and the plate rubber bearing will undergo various degrees of frictional slip under seismic force with the occurrence of shear deformation of the plate rubber bearing. In this case, the force and displacement relationship of the bearing satisfies the typical bilinear constitutive relations, and the seismic force of the bearing is constant after the sliding of bearings based on the fundamental hypothesis.

Therefore, the slip energy dissipation capacity of the bearings in the horizontal and vertical directions was simulated by the Combin40 spring element with the bilinear strain-hardening model and the influence of vicious damper to fully simulate the actual status of the bridge. The mechanical principle of the diagram is shown in Figure 19.

The real constant of this element requires five parameters which are, respectively, K1 (spring constant of spring 1), C (damping constant), M (mass), GAP (gap size), FSLIDE (limit sliding force), and K2 (spring constant of spring 2). Meanwhile, the actual effect of seismic isolation of the bearings was simulated through inputting and adjusting the five parameters, and the spring stiffness in all directions is selected according to the bearing parameters set by the prototype bridge.

Therefore, the plate rubber bearing in the finite element model of this paper consists of three elements to simulate all three of its directions, respectively. In order to achieve the effect of approximate simulation, all of the degrees of freedom of disjointed nodes of three elements will be constrained, and the rotational degree of freedom of the nodes at the intersection are constrained.

The main structure of the new alternative retainers studied in this paper, being different from the traditional concrete blocks, is made of steel, and it is a fabricated structure in which its connection between the components relies on high-strength bolts or welding.

#### 4.2.3. Loading Protocols

Figure 20 shows three seismic acceleration time-history curves. Based on the seismic fortification theory of “three level” of the transverse bridge, the ground motion acceleration amplitude was adjusted, respectively, to 0.05 g (6 degrees of seismic fortification intensity), 0.1 g (7 degrees of seismic fortification intensity), 0.2 g (8 degrees of seismic fortification intensity), 0.3 g (8 degrees of semi-seismic fortification intensity) and 0.4 g (9 degrees of seismic fortification intensity), and was input in return.

### 4.3. Analysis of Seismic Performance of Transverse Bridge

#### 4.3.1. Displacement Response of the Girder

Due to the limited space, only the time-history diagram of the 0.1 g working condition is listed, as shown in Figure 21 to Figure 22. When the ground motion acceleration peak is 0.05 g–0.1 g, the peak value of transverse absolute displacement of the girder under the El-Centro seismic wave reaches the maximum without a retainer, which is increased by 3.43 mm (22.35%, 6.88 mm (22.42%) compared with that under the Imperial valley seismic wave, and increased by 6.97 mm (59.02%) and 13.94 mm (58.99%) compared to the Taft seismic wave. The peak value of transverse absolute displacement of the girder under the El-Centro seismic wave remains the maximum, however, when the peak value is 0.05 g, the value is decreased by 1.18 mm (8.93%) under the El-Centro seismic wave and the Imperial valley seismic wave, and the subtraction of the peak displacement decreased by 2.52 mm (22.89%) compared with the Taft seismic wave. When the peak value is 0.1 g, the subtraction of the peak displacement of transverse absolute displacement of the girder continues to decrease under different ground motion spectrum characteristics, such as the El-Centro seismic wave reducing by 2.39 mm (9.05%) and 5.05 mm (22.93%), respectively, compared with the Imperial valley seismic wave and the Taft seismic wave.

It can be seen that the El-Centro seismic wave has an amplification effect on transverse absolute displacement of the girder, and it also reflected between the Imperial valley seismic wave and the Taft seismic wave that is the transverse absolute displacement of the girder of the Imperial valley seismic wave average increased by 29.93% (without the retainers), 19.98% (with the new alternative retainer) compared with the Taft seismic wave.

However, the data show that the bridge with the new alternative retainer has a certain inhibitory effect on the amplification effect. The displacement response of the main girder of the bridge structure with the new alternative retainer is more balanced under the different seismic frequency spectrum characteristics compared to that without retainers, which prompts the ground motion spectrum adaptability of bridge structures.

As shown in Figure 22, after the bridge adopts the new retainers, the absolute peak displacement of the girder average reduces by 13.93% of the Imperial valley seismic wave, reduces by 17.96% of the El-Centro seismic wave, and reduces by 6.96% of the Taft seismic wave. In addition, the percentage of absolute peak displacement of the girder with new retainer is basically the same under different ground motion intensities, for example, the deduction in the percentage reduction in the peak displacement between different ground motion intensities is only 0.87%.

The data shows that absolute peak displacement of the girder with new retainer average decrease by 12.95% compared without retainers. The new retainer has excellent capacity of limiting displacement that does not fluctuate violently due to the change of ground motion intensity and showing a stable situation under the same ground motion spectrum characteristics. The reason is that when the earthquake occurs, sliding occurs at the bottom of the girders and at the plate rubber bearing, and when there are no retainers, because the bearing and the main girders are not constrained and the inertial force cannot be transmitted to the pier, the bearing was damaged leading to the large displacement of the main girders. When a rare earthquake occurs, the new alternative retainers with a large deformation range and good energy dissipation capacity can restrain the displacement of the bearing and the girders, avoiding rapid loss of function, which can maximize the transmission of inertial force and impose a limiting function.

#### 4.3.2. The Shear Response of the Pier Bottom

As shown in Figure 23, When the ground motion acceleration peak is 0.05 g–0.4 g, at the same peak value, the peak absolute shear force at pier bottom under the El-Centro seismic wave is increased by an average of 43.19% (without the retainers) and 38.00% (with the new alternative retainers) compared with the Imperial valley seismic wave, and increased by an average of 32.70% (without the retainers), 39.00% (with the new alternative retainers) compared with the Taft seismic wave: the average increase in the absolute shear force peak value at the pier bottom under the Imperial valley seismic wave is 8.06% (without the retainers) and -0.50% (with the new alternative retainers), respectively, than that of the Taft seismic wave. After installing of the new retainers, the peak absolute shear force at the bottom of the pier under the Imperial valley seismic wave is increased by an average of 11.28%, increased by an average of 7.08% of the El-Centro seismic wave, and increased by an average of 2.40% of the Taft seismic wave.

The data shows that the peak absolute shear force at the bottom of the pier with the new retainers presents an increasing trend, and is increased by an average of 6.92%. Meanwhile, the peak absolute shear force at the bottom of the pier with the new retainers under different seismic motion spectrum characteristics will be enlarged or reduced.

#### 4.3.3. The Bending Moment Response of Pier Bottom

As shown in Figure 24, when the ground motion acceleration peak is 0.05 g–0.4 g, at the same peak value, the peak absolute value of the bending moment at the pier bottom under the El-Centro seismic wave is increased by an average of 35.23% (without the retainers) and 30.55% (with the new alternative retainers) compared with the Imperial valley seismic wave, and increased by an average of 25.51% (without the retainers), and 30.88% (the new alternative retainers) compared with the Taft seismic wave. The peak absolute value of the bending moment at the pier bottom under the Taft seismic wave is increased by an average of 7.68% (without the retainers) compared with the Imperial valley seismic wave. After installing the new retainers, the trend was reversed, in that the value of the pier under the Imperial valley seismic wave increased by an average of 11.28% (with the new alternative retainers) compared with the Taft seismic wave. That is because installing the new alternative retainers causes that the average increase in the absolute bending moment peak value at the pier bottom to be 10.60% of the Imperial valley seismic wave and 6.73% of the El-Centro seismic wave. The average increase rate of the value of the bending moment increase at the pier bottom under the Imperial valley seismic wave exceeds that of the Taft seismic wave.

The data show that the peak absolute value of the bending moment at the pier bottom with the new retainers presents an increasing trend, and is increased by an average of 6.92%. Meanwhile, the ranking relationship of peak absolute value of the bending moment at the pier bottom with the new retainers under different seismic motion spectrum characteristics will be reversed due to different enlargement ratios. When the earthquake occurs, the reason is that the transverse stiffness of the bridge with the new alternative retainer is increased, leading to the inertial force being transmitted to the bridge pier when the main girder is displaced, which inevitably increases the pier seismic response. However, the seismic response of the pier is not greatly increased because of the working mechanism of energy dissipation in stages of the new retainer and its good energy dissipation capacity.

## 5. Conclusions

The paper studies the seismic performance of an alternative retainer that focuses on three functions of “limiting displacement”, “energy consumption”, and “alternative link”. The results show that:(1)The failure mode of the new alternative retainer in the test is consistent with the design working mechanism. The two-stage graded energy dissipation of the alternative link compared to the one-time slant shear failure of the traditional concrete block, which prolongs the life cycle of the retainer, mitigates the damage of the bent cap, and is convenient to retrofit and reinforcement both retainer and bridge post-earthquake.(2)Transverse replacement of the main girder with the new alternative retainers is greatly reduced compared to without retainers. Although the seismic response of the pier inevitably increases, its increasing is gentle because the excellent energy consumption of the new retainer causes the minor inertia force to be transmitted to the substructure while maintaining the limiting displacement effect.(3)The key index affecting the seismic performance of the new alternative retainers is the thickness of the web of the alternative link, however, the influence of other design parameters, such as the thickness of the vertical plate and the number of alternative links, on the seismic performance of the new retainer will require further research.

## Figures and Tables

**Figure 1 polymers-14-03506-f001:**
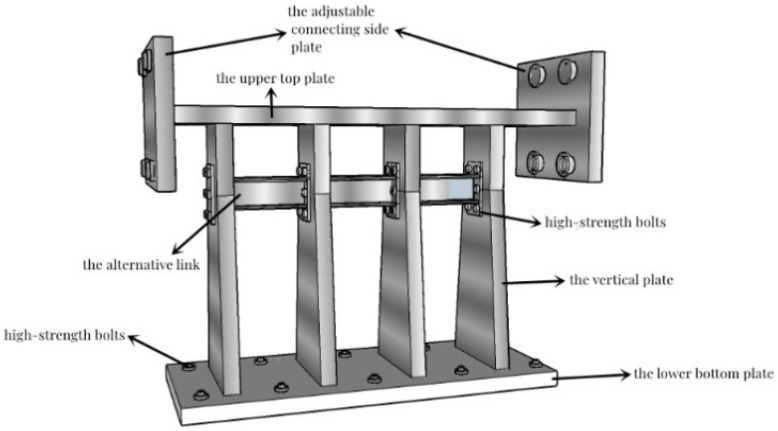
The new alternative retainer.

**Figure 2 polymers-14-03506-f002:**
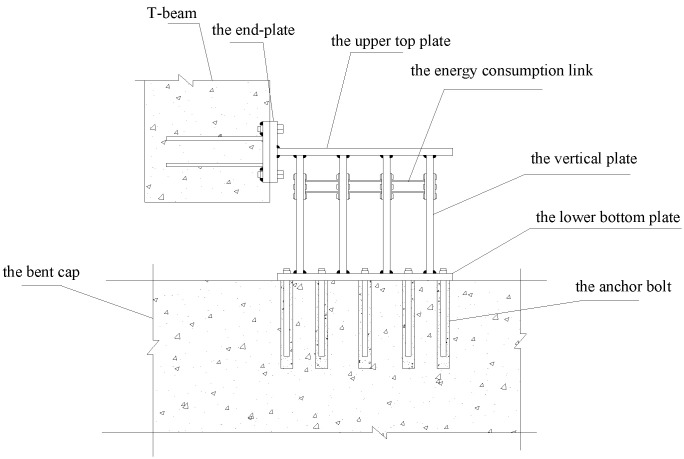
Installation diagram.

**Figure 3 polymers-14-03506-f003:**
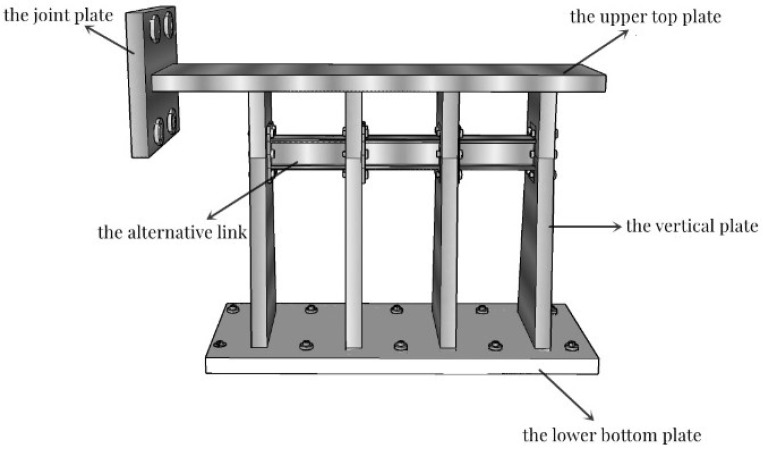
The structure design of test specimens.

**Figure 4 polymers-14-03506-f004:**
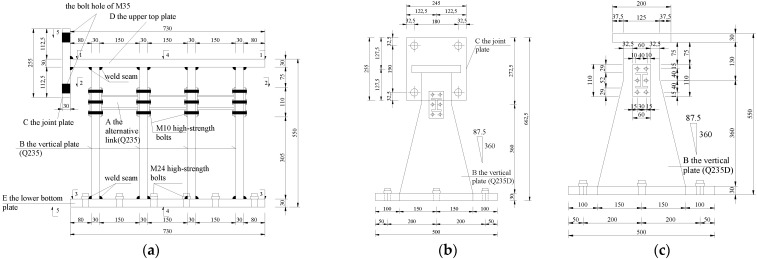
Specimen size drawing. (**a**) Standard specimen elevation; (**b**) standard specimen vertical plate; (**c**) standard specimen loading end-plate; (**d**) the alternative link plan view; (**e**) the alternative link elevation view; (**f**) the alternative link side view.

**Figure 5 polymers-14-03506-f005:**
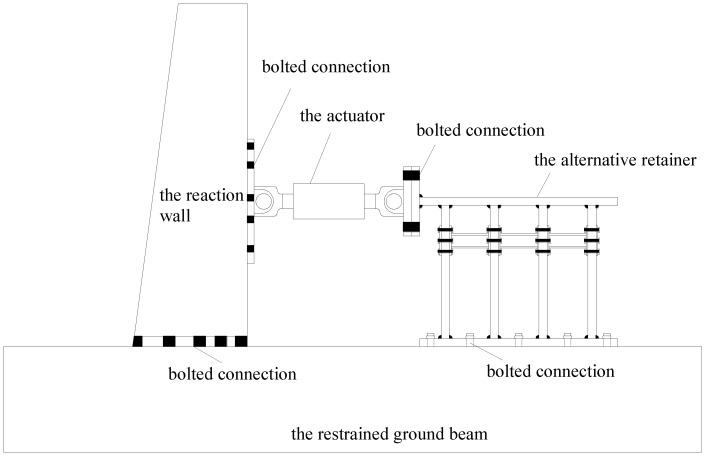
Test setup.

**Figure 6 polymers-14-03506-f006:**
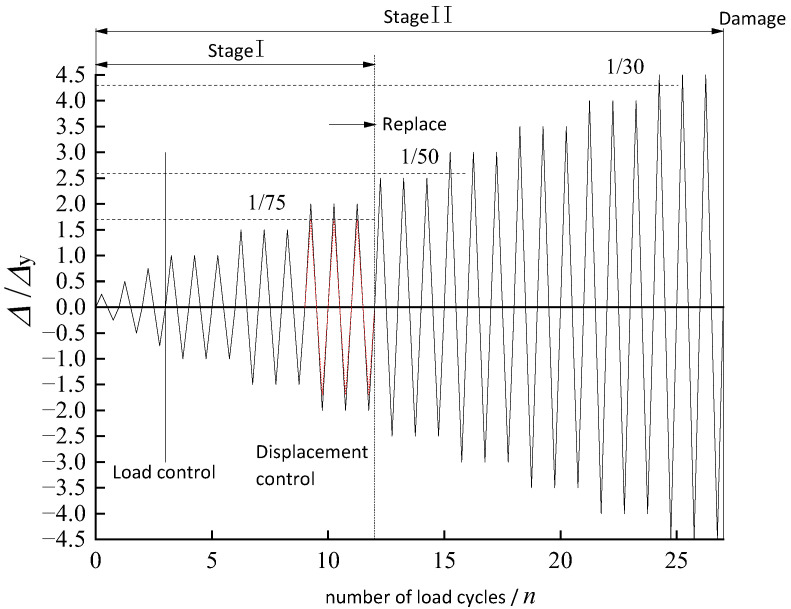
Loading protocols.

**Figure 7 polymers-14-03506-f007:**
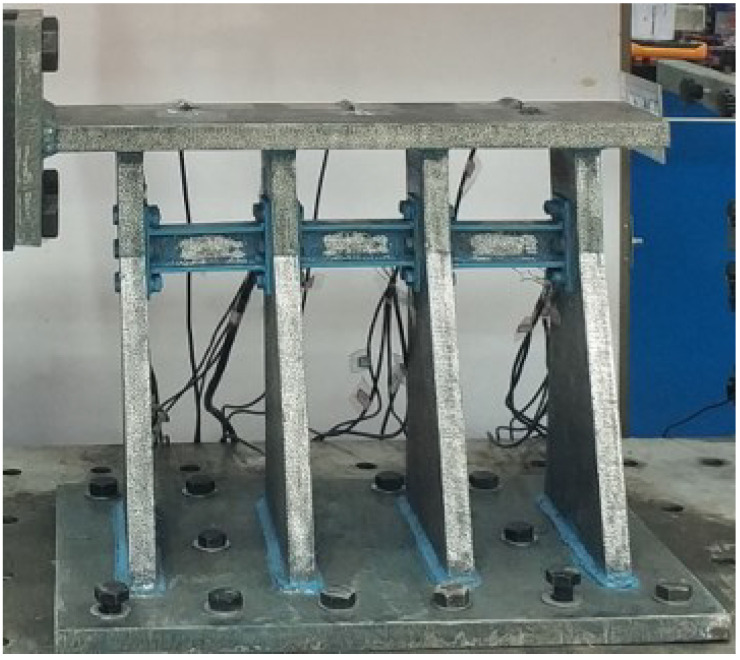
Deformation of the specimen during loading.

**Figure 8 polymers-14-03506-f008:**
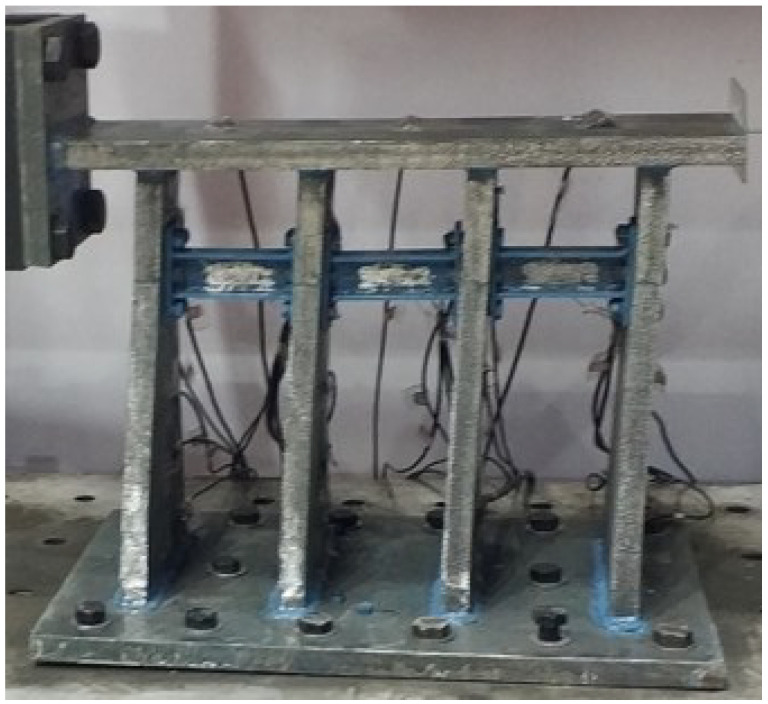
Specimen deformation Δ = 80 mm.

**Figure 9 polymers-14-03506-f009:**
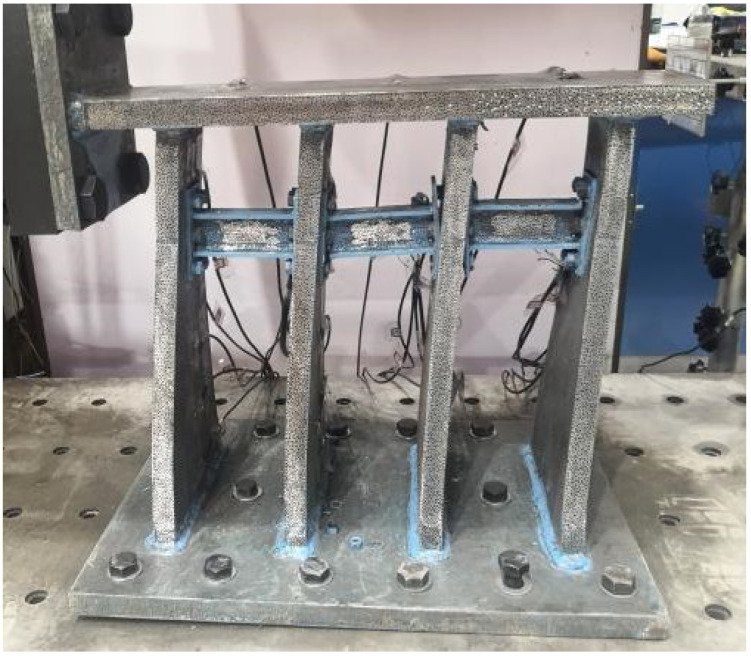
Deformation of the specimen Δ = 100 mm.

**Figure 10 polymers-14-03506-f010:**
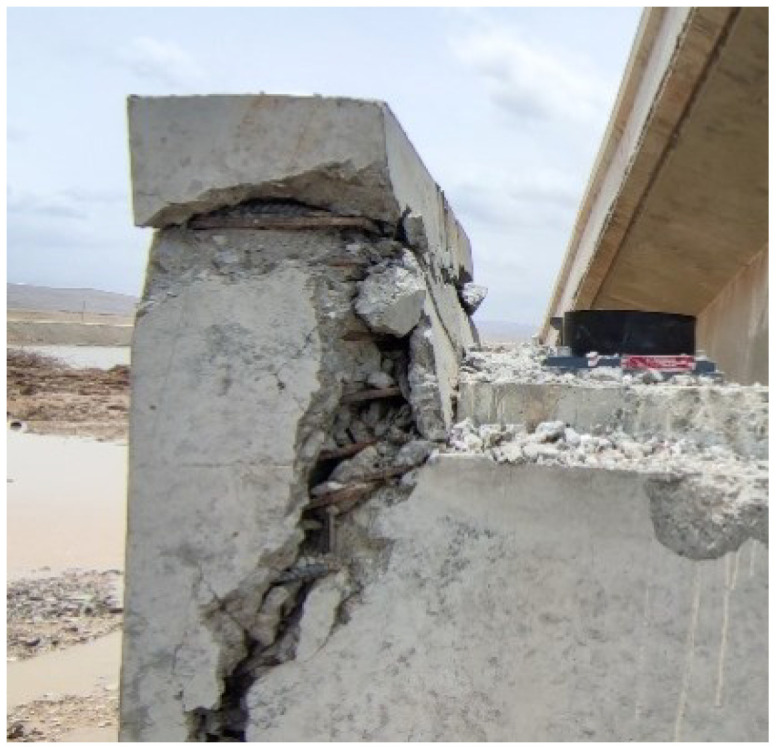
Seismic damage of blocks.

**Figure 11 polymers-14-03506-f011:**
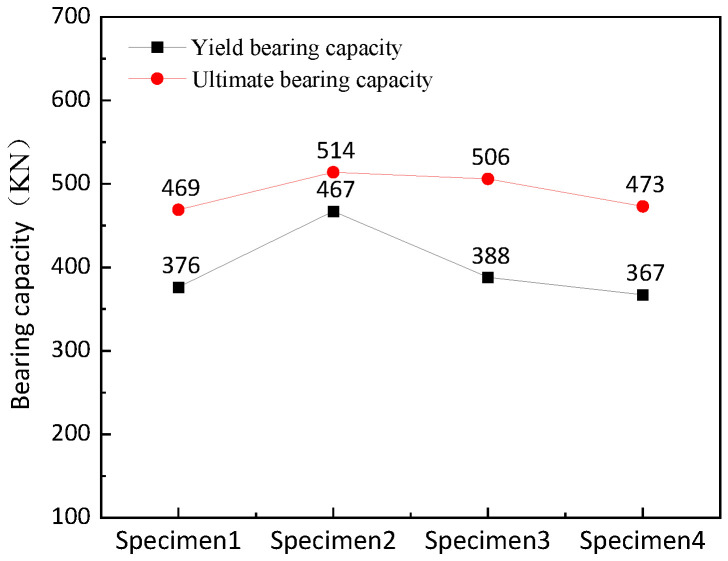
Comparison of bearing capacity of specimens.

**Figure 12 polymers-14-03506-f012:**
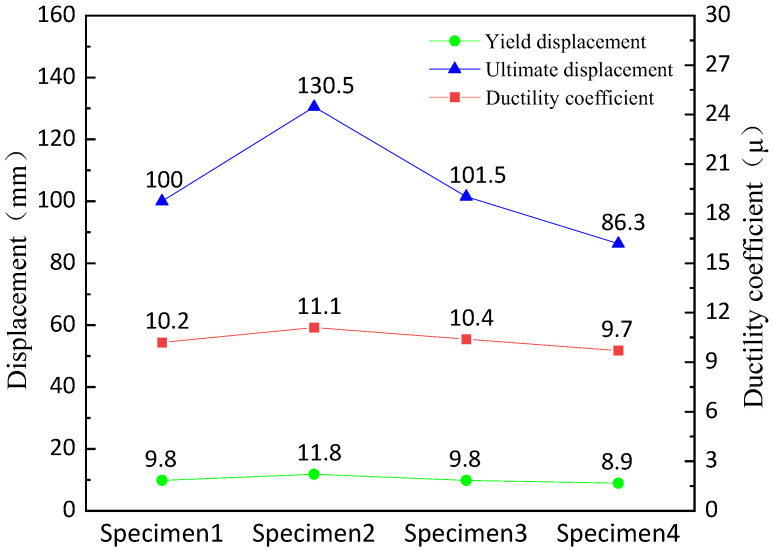
Comparison of displacement and ductility of each specimen.

**Figure 13 polymers-14-03506-f013:**
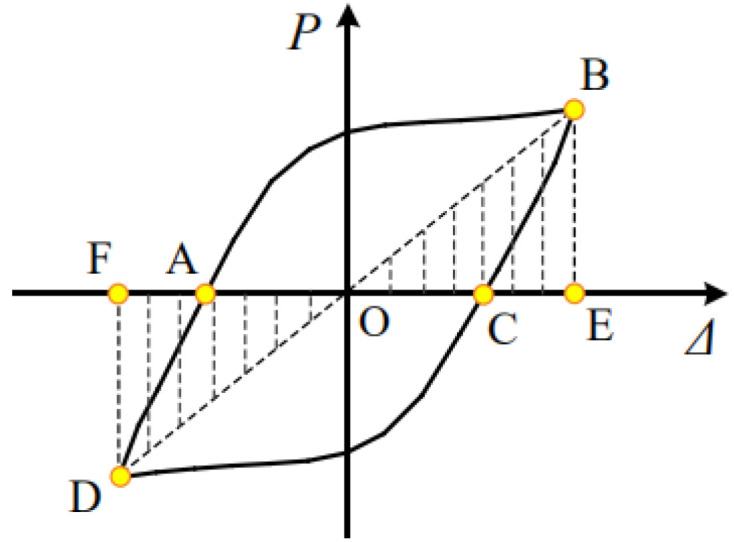
Force–displacement hysteretic curves.

**Figure 14 polymers-14-03506-f014:**
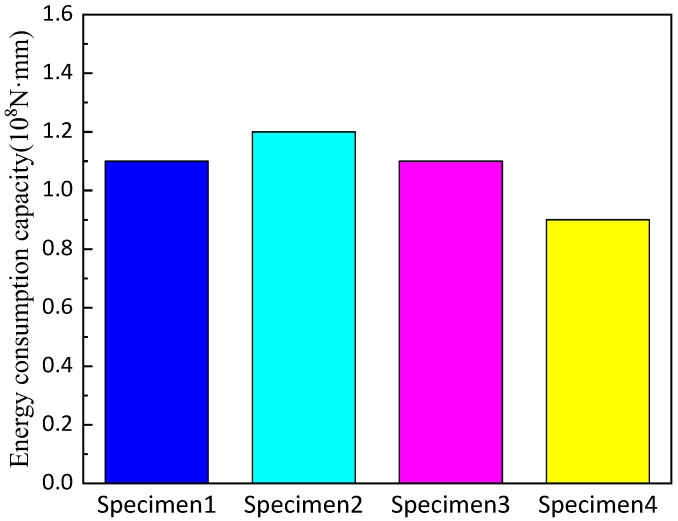
Comparison of energy consumption capacity of the specimens.

**Figure 15 polymers-14-03506-f015:**
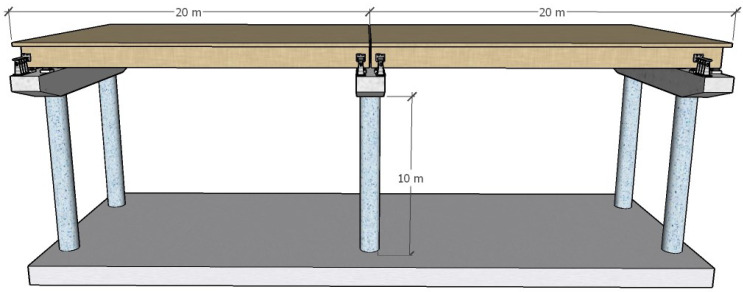
Bridge prototype.

**Figure 16 polymers-14-03506-f016:**
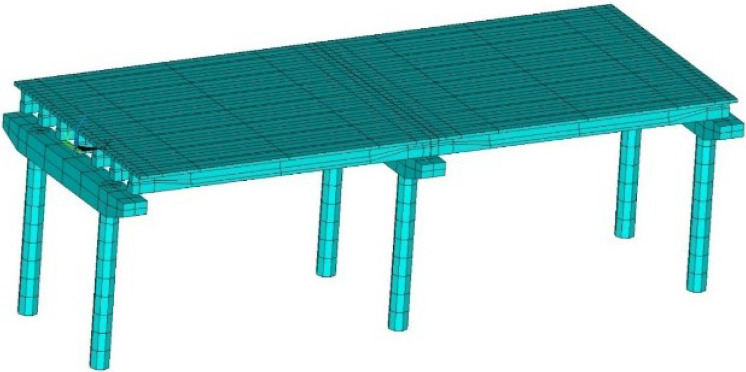
The finite element model of prototype bridge of shaking table test.

**Figure 17 polymers-14-03506-f017:**
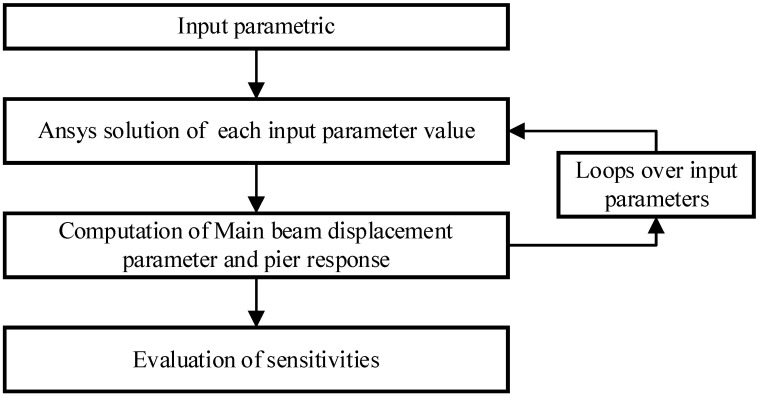
The flow-process diagram of filtering the sensitive parameters.

**Figure 18 polymers-14-03506-f018:**
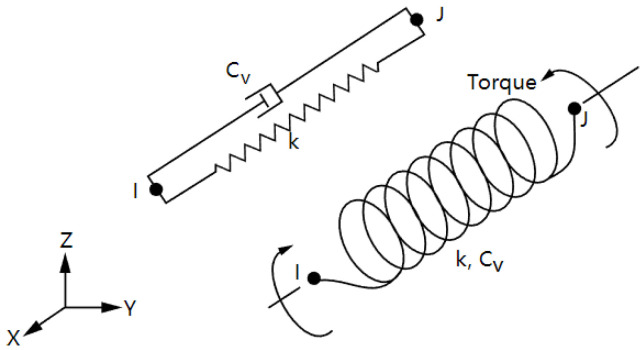
The mechanical principle of diagram of Combin14 spring element.

**Figure 19 polymers-14-03506-f019:**
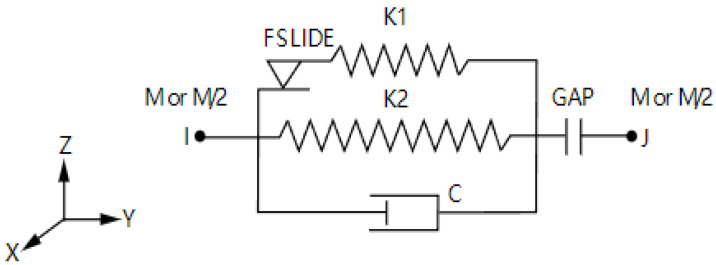
The mechanical principle of the diagram of Combin40 spring element.

**Figure 20 polymers-14-03506-f020:**
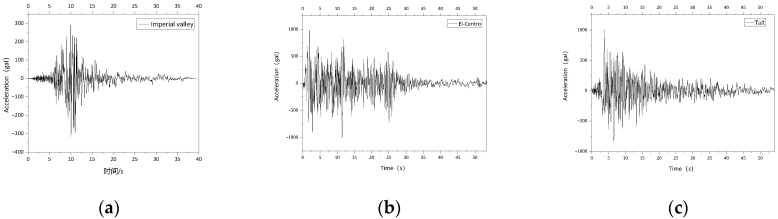
Seismic wave acceleration time-history curve. (**a**) Imperial valley seismic wave; (**b**) El-Centro seismic wave; (**c**) Taft seismic wave.

**Figure 21 polymers-14-03506-f021:**
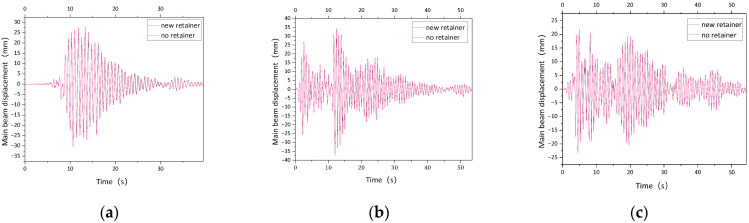
Time-history contradistinction of lateral displacement of main girder (0.1 g).(**a**) Imperial valley seismic wave; (**b**) El-Centro seismic wave; (**c**) Taft seismic wave.

**Figure 22 polymers-14-03506-f022:**
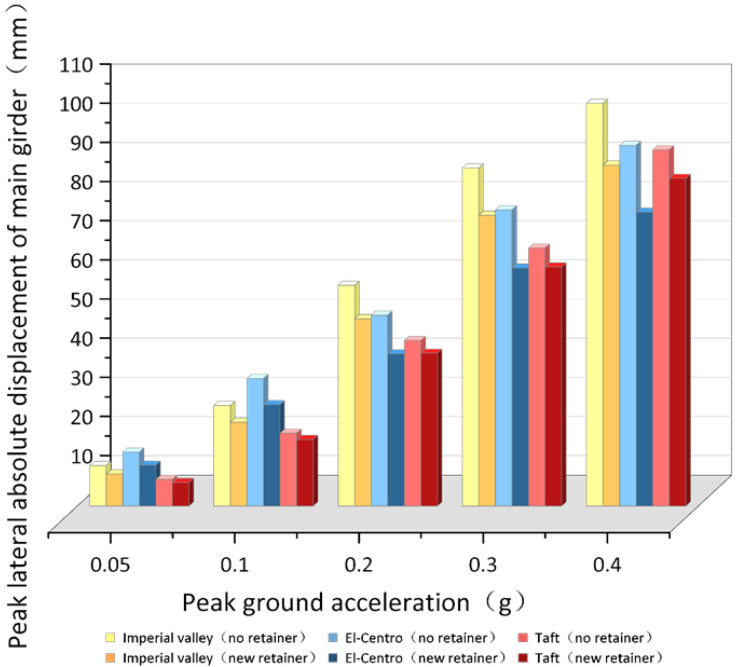
Contradistinction of the peak value of transverse absolute displacement of the girder.

**Figure 23 polymers-14-03506-f023:**
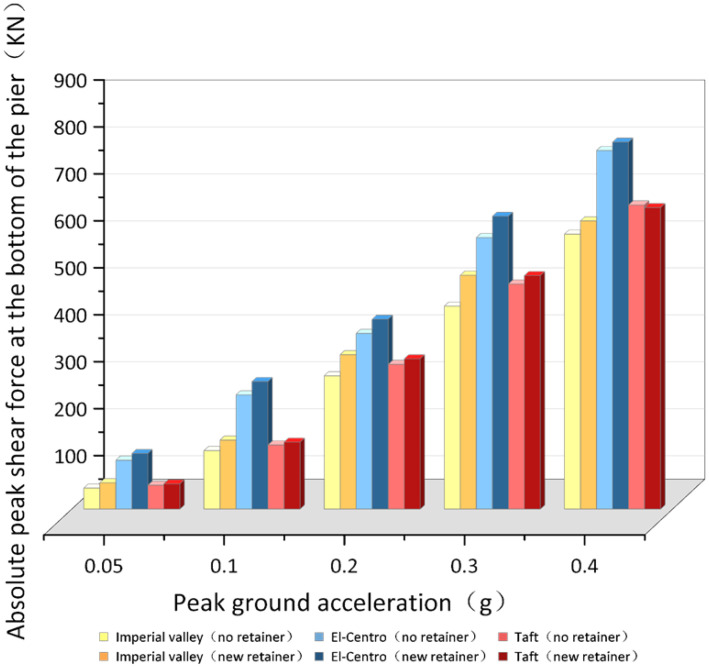
Contradistinction of the peak absolute shear force at pier bottom.

**Figure 24 polymers-14-03506-f024:**
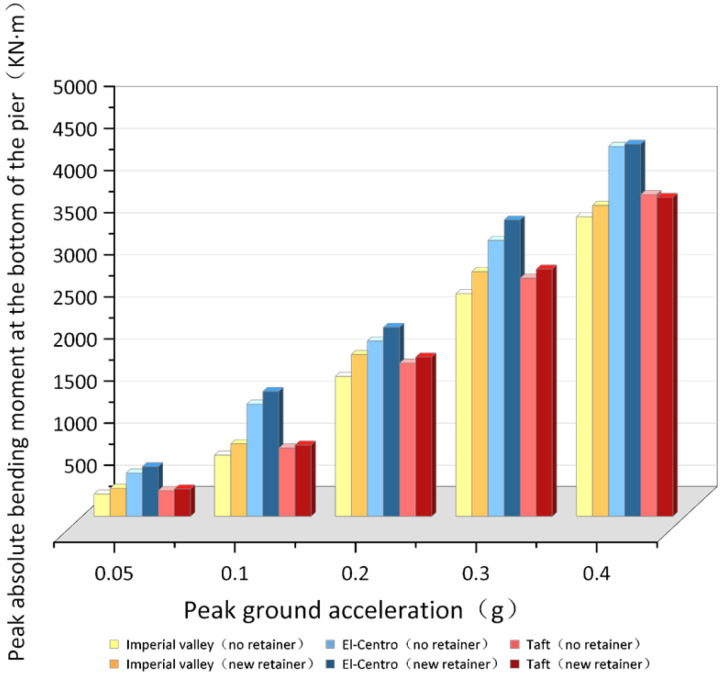
Contradistinction of the peak absolute value of bending moment at pier bottom.

**Table 1 polymers-14-03506-t001:** Dimension parameters of scaled specimens.

Specimen Number	The Alternative Link Height (mm)	Web Thickness (mm)	The Location of Alternative Link (mm)
1	52(6 + 40 + 6)	6	360
2	52(6 + 40 + 6)	8	360
3	62(6 + 40 + 6)	6	360
4	52(6 + 40 + 6)	6	400

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
