# Peer review of "Seismic Performance and Engineering Application Investigation of a New Alternative Retainer"

_polymers, 2022, doi:10.3390/polym14173506_

Round 1

Reviewer 1 Report

This article was inspired by the failure during the Wenchuan earthquake. A large number of plate rubber bearings were there completely destroyed.

Athors prepare structure design of new retainer and done a the quasi-static test.

The loading protocols and failure criterion are important in this experiment, which the Authors did.

The work contains important conclusions that result from the performance of the experiment.

Tests on samples were performed, followed by a bridge model analysis.

The experience and conclusions obtained on their basis also apply to other similarly designed bridges.

Conclusions are short, but substantively good.

References are also good, although they include studies from America and Europe to a small extent.

In general, the paper is acceptable after minor revision, as below.

Remarks:

1. In my opinion, chapter 1.1 should be the first in chapter 2.

2. There is a typographical error in the title of chapter 3: "Experiment analysis4 (?)"

Author Response

Dear Reviewer,
Thanks very much for taking your time to review this manuscript. Please see the attachment for our reply.

Reviewer 2 Report

This manuscript concerns an idea of some structural damper necessary for the structures resting on the subsoil possibly subjected to the earthquake waves. This idea is generally interesting, but its realization needs substantial modifications. Language and editing need remarkable changes (cf. "As shown in Figure 22, When the ground motion acceleration peak (...) as the example)". English style is very schematic and skimpy, so the Authors are encouraged to consult this elaboration with a native or professional English editing company. 

Secondly, the Finite Element Method analysis description is very imprecise - discretization way has been postponed (a number of the FEs etc.) together with the solution algorithm and necessary computer power. The Authors may follow a content of the paper: Ł. Figiel, M. Kamiński, Numerical probabilistic approach to sensitivity analysis in a fatigue delamination problem of a two layer composite. Applied Mathematics and Computation 209(1): 75-90, 2009. 

This work can be re-evaluated after these modifications for possible publication in this journal. 

Author Response

(The authors gave the same response as above.)

Round 2

Reviewer 2 Report

The manuscript has been improved by the Authors and now it is ready for publication in this journal.